# X-ClusterLink: An Efficient Cross-Cluster Communication Framework in Multi-Kubernetes Clusters

Submission Id: 352

## Abstract

Kubernetes is widely adopted by enterprises to enhance service availability for applications such as web services and large-scale model training, due to its advantages in managing containerized applications. As service demands increase, a single Kubernetes cluster often becomes insufficient, leading to the trend of using multiple clusters to improve service scalability. However, achieving efficient cross-cluster communication poses significant challenges due to the need for low latency, high throughput, and strong robustness. Existing methods for cross-cluster communication either employ a centralized control plane, which becomes a communication bottleneck, or use numerous service-bound proxies, leading to increased management complexity and possibly compromised robustness in cross-cluster communication.

To address the above challenges, we introduce X-ClusterLink, a framework designed for efficient cross-cluster communication in multi-Kubernetes clusters. X-ClusterLink first employs broker clusters to ensure low-latency cross-cluster synchronization. Then, it aggregates multiple containerized gateways to enhance throughput and leverages eXpress Data Path (XDP) for advanced packet processing, thereby accelerating traffic forwarding. Finally, it incorporates Bucket-Based Consistent ECMP to facilitate seamless failover and enhance robustness. Experimental results demonstrate that X-ClusterLink significantly improves cross-cluster communication efficiency, increasing cross-cluster forwarding bandwidth by $3.1 \times$ compared to existing solutions.

## Keywords

Virtualization, Resource Management, Kubernetes, Traffic Forwarding, Web Infrastructure

## 1 Introduction

In the contemporary digital landscape, a growing number of web applications are hosted on cloud servers to provide cost-effective services. Enterprises increasingly seek automated, scalable, and highly dependable management systems for these applications to enhance operational efficiency and service quality. In response to these demands, many enterprises advocate virtualization technologies for managing web applications [5]. Kubernetes [3, 4], a premier open-source container orchestration platform, excels in automating the deployment, scaling, and management of containerized applications. Specifically, its flexibility and scalability make it particularly suitable for businesses that require robust resource management in their web systems [21]. Additionally, Kubernetes is increasingly utilized for flexible task scheduling in large-scale model training, effectively improving resource management and making workflows more efficient for fluctuating workloads. [17, 24]. Consequently, a growing number of enterprises are beginning to adopt Kubernetes to manage services.

Typically, Kubernetes manages a set of **nodes**, which are either physical or virtual machines, in a unified manner to provide services to users. These nodes collectively form what is known as a Kubernetes **cluster** [4]. Each node can host multiple **pods**, which are the fundamental operational units within Kubernetes. Typically, a **pod** is a container running specific applications, such as web services, databases, or model training jobs. To ensure high availability, Kubernetes can deploy identical pods across different nodes, providing consistent **service** [4]. This setup maintains service continuity if a node fails and allows efficient load balancing through automatic scaling based on demand and resource usage.

With escalating service demands, a single Kubernetes cluster frequently becomes insufficient, prompting the adoption of multiple clusters to enhance scalability. Specifically, a standard Kubernetes cluster only supports up to 5,000 nodes and 150,000 pods. However, large tenants may require a private cloud that hosts millions of containers, far surpassing the capacity of a single cluster [28]. Additionally, deploying multiple Kubernetes clusters not only boosts fault tolerance and system availability through redundancy and isolation mechanisms but also facilitates the implementation of comprehensive security measures, substantially mitigating the risks of lateral attacks and unauthorized access [34].

However, effective communication across multiple Kubernetes clusters entails three critical requirements [8]. (1) *Low Latency*: The substantial data synchronization traffic between clusters is highly sensitive to latency, necessitating low-latency, real-time communication to effectively manage sudden spikes. [14]. For example, a global streaming service employs multiple Kubernetes clusters to serve users across regions, requiring real-time data synchronization to ensure data consistency at low latency [16]. (2) *High Throughput*: High data transmission rates are crucial for forwarding large traffic volumes between clusters, particularly in environments where large models are trained on multiple Kubernetes clusters. Thus, designing high throughput traffic forwarding solutions can considerably reduce data processing times and improve the efficiency of model training [17, 24]. (3) *Strong Robustness*: Although individual components generally exhibit low failure rates, large-scale cloud environments remain vulnerable to network anomalies, such as node failures [6, 26] and traffic surges [30, 36]. In scenarios such as financial transactions on Kubernetes clusters, maintaining a robust network with fast failover is essential to prevent disruptions and ensure robust service.

Existing solutions for cross-cluster communication, such as Kubernetes Federation [15] and Istio Service Mesh [13], unfortunately do not fully meet the above requirements. Specifically, Kubernetes Federation orchestrates cross-cluster service communications using a central control plane, whereas Istio Service Mesh employs Envoy proxies for each service to facilitate cross-cluster communication. On the one hand, the centralized management structure of the Kubernetes Federation requires that all cross-cluster communications

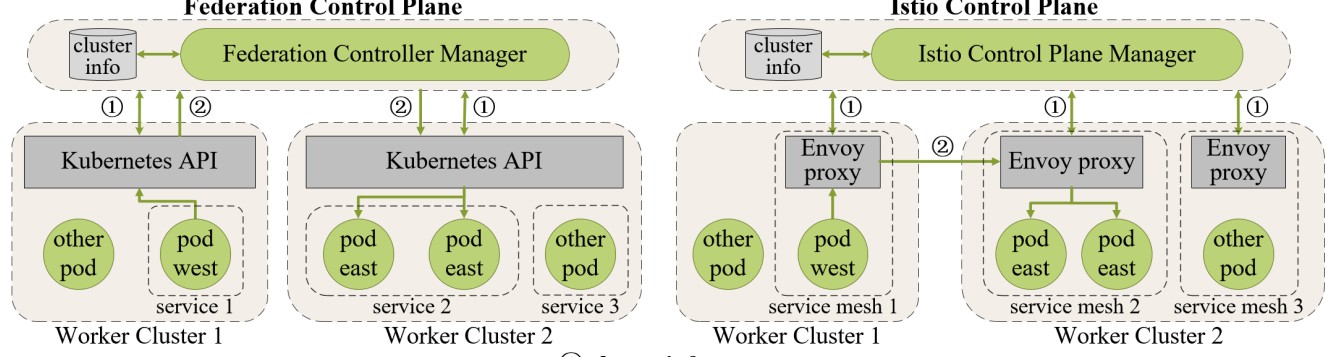

Figure 1: Two Typical Cross-Cluster Communication Solutions for Multi-Kubernetes Clusters: (a) Kubernetes Federation establishes a Control Plane to manage service communications across multiple clusters. (b) Istio Service Mesh creates Envoy proxies in each service, then communicates across different clusters by Envoy proxies.

pass through a single control plane. This central control plane can become a significant bottleneck, especially during periods of burst cross-cluster traffic, consequently limiting system throughput and elevating latency, thereby adversely affecting overall performance and responsiveness. On the other hand, the deployment of numerous service proxies by Istio Service Mesh technologies complicates management and introduces an additional layer, increasing the risk of configuration errors and reducing system robustness.

To overcome the above challenges, we propose X-ClusterLink, a framework for cross-cluster communication in multi-Kubernetes clusters. Specifically, X-ClusterLink includes the following innovative designs: (1) X-ClusterLink implements a broker cluster (§4.1) and the Rounding-Based Broker Cluster Mapping (RBCM) algorithm (§4.2) to manage cross-cluster communication, incorporating built-in load-balancing features to ensure low latency in cross-cluster information synchronization. (2) X-ClusterLink enhances cross-cluster network throughput by deploying multiple containerized gateway aggregations. It also accelerates traffic forwarding by using extended Berkeley Packet Filter (eBPF) and Express Data Path (XDP) [10], an advanced packet processing technology, within these gateways [31]. (§5.1) (3) X-ClusterLink achieves fast failover through the proposed Bucket-Based Consistent ECMP [12], thereby enhancing system robustness. (§5.2)

The principal contributions of this paper are outlined as follows:

(1) We conduct a detailed analysis of the advantages and disadvantages of existing typical solutions for cross-cluster communication in multi-Kubernetes clusters, and subsequently present the design goals of our proposed framework.
(2) We design X-ClusterLink, a prototype framework comprising the Broker Layer and the Worker Layer, to achieve efficient cross-cluster communication in multi-Kubernetes clusters. We plan to open-source it at Github soon.
(3) We evaluate the efficiency of X-ClusterLink through testbed experiments. The results show that X-ClusterLink significantly reduces the impact of burst traffic and abnormalities on performance. Notably, X-ClusterLink increases cross-cluster forwarding bandwidth by 3.1 × compared to state-of-the-art solutions.

## 2 Background and Preliminaries

### 2.1 Multi-Kubernetes Clusters Communication

Kubernetes, released by Google in 2014 and derived from their internal system Borg used for apps like Gmail, simplifies the deployment, scaling, and management of containerized applications across multiple nodes [35]. It enhances efficiency and adaptability, leading to its widespread use in managing complex web systems and supporting large-scale model training [17, 21, 24].

Kubernetes manages multiple **nodes**, both physical and virtual, to deploy, scale, and manage applications efficiently. These managed nodes are collectively known as a **cluster** [4]. Each node supports several pods, the essential operational units within Kubernetes. Typically, a **pod** contains a container that runs specific applications, such as web services, databases, or model training programs. To maintain high availability, Kubernetes distributes identical pods across different nodes to ensure consistent **service** [4].

Deploying multi-Kubernetes clusters is crucial in modern cloud-native architectures, enhancing the scalability and fault tolerance of web applications while facilitating service deployment across cloud platforms [28]. This arrangement optimizes resource utilization and improves load balancing, significantly reducing single points of failure and boosting system reliability. Moreover, using multiple clusters for large-scale model training distributes computational loads and data storage demands, effectively addressing the scalability constraints of a single cluster [34].

Cross-cluster communication is vital in multi-cluster environments, enabling interaction between services across different clusters, such as financial services, to ensure data consistency [8]. It facilitates rapid information synchronization, maintaining uniform service delivery across various clusters. Moreover, when using multiple clusters for large-scale model training, high-throughput cross-cluster communication improves training efficiency, keeps the system performing optimally under heavy loads, and enhances the user experience [17].

### 2.2 Limitations of Prior Works

Istio Service Mesh [7, 13] is a widely utilized tool that enhances cross-cluster communication by deploying Envoy proxies for each

service and integrating them into Istio's service system, thereby improving network functionality [18]. This configuration enables seamless communication across services located in different clusters. The Istio control plane manages synchronization and routing rules among Envoy proxies to ensure smooth cross-cluster connectivity. However, since Istio operates on a per-service basis, each service requiring cross-cluster communication necessitates its own Envoy proxy. This requirement leads to an increase in the number of proxies when managing multiple cross-cluster services, potentially degrading performance. Furthermore, the proliferation of proxies complicates management, increases the overhead for information synchronization, and heightens the risk of configuration errors, which may result in communication failures [37].

Skupper [32] connects multiple Kubernetes clusters at the application layer (Layer 7), creating a temporary virtual network for secure cross-cluster communication. It utilizes the Virtual Application Network (VAN) to link applications and services within a hybrid cloud, effectively simulating co-location by operating at Layer 7. Application routers manage communication between addresses. However, Skupper's operation at this higher layer results in performance overhead and increased resource consumption [1].

Submariner [33] improves the connectivity of Kubernetes clusters by establishing secure IPSec tunnels. Each cluster employs a Submariner Gateway Pod for tunnel management and the Lighthouse component to facilitate cross-cluster communication, thereby enhancing service connectivity. However, its reliance on a single cross-cluster gateway means that Submariner cannot provide seamless failover during gateway failures, critically and significantly compromising its robustness.

Despite the capabilities of existing solutions to communication across multiple clusters, they present several inherent challenges. These include: (1) *Significant Additional Latency:* In large-scale deployments, additional control planes can increase the latency of information synchronization between clusters by creating bottlenecks, leading to data inconsistencies that may affect critical services like finance. (2) *Limited Cross-Cluster Throughput:* Existing multi-cluster architectures often underutilize the network bandwidth of each node, leading to insufficient throughput in high-demand scenarios. This limitation impedes the effective transfer of large data volumes between clusters, severely affecting performance-sensitive applications. (3) *Insufficient Fault Resilience:* Current multi-cluster solutions often depend on a single gateway to forward cross-cluster communication, which inadequately handles network failures or peak traffic. As a result, cross-cluster communication is prone to disruptions, resulting in a lack of robustness.

## 3 System Design
### 3.1 Design Goals
X-ClusterLink is a framework for cross-cluster communication in multi-Kubernetes clusters. Our design goals are as follows:

- **Low Latency**: Low-latency cross-cluster communication is essential for maintaining information synchronization and data consistency across clusters. X-ClusterLink employs broker clusters and load-balancing algorithms to manage burst traffic and minimize delays.

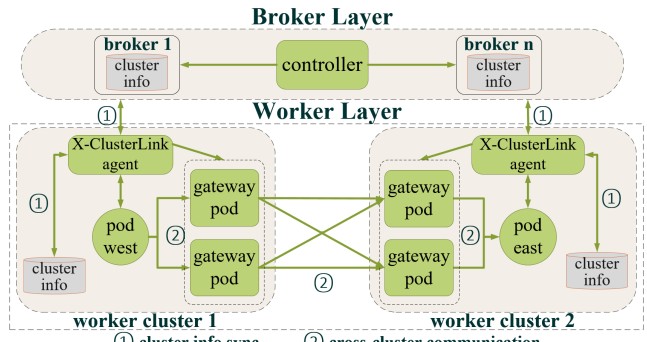

**Figure 2: X-ClusterLink Overview:** The Broker Layer manages traffic forwarding between clusters and ensures synchronization of cluster information. The Worker Layer focuses on providing efficient, large-scale cross-cluster communication while ensuring failover and load balancing among the cluster's gateways.

- **High Throughout**: The framework needs to quickly forward large amounts of cross-cluster data to support high-performance applications spanning multiple clusters. X-ClusterLink aims to aggregate the bandwidth of various nodes within each cluster to meet this demand.
- **Fault Transparency**: High reliability is crucial for cross-cluster communication, requiring continuous service availability even during failures. Therefore, X-ClusterLink is designed to ensure seamless failover, making disruptions transparent to services.
- **Scalability**: With the rapid growth of demand, the framework must scale without incurring excessive overhead. X-ClusterLink needs to lower the overhead of syncing cross-cluster information to support large-scale deployments and efficiently expand as required.

### 3.2 System Overview
As shown in Figure 2, X-ClusterLink achieves the above design goals through two main components: the **Broker Layer** and the **Worker Layer**. Below is a brief overview of these two layers.

**Broker Layer** effectively manages traffic forwarding between clusters and ensures synchronization of information across them. This layer utilizes multiple brokers to achieve low-latency synchronization and is adept at supporting effective load balancing. Each broker manages and directs cross-cluster traffic, ensuring accurate delivery of data packets to their intended destinations. This functionality is enabled by providing an interface for cross-cluster communication within each cluster [38].

**Worker Layer** focuses on providing efficient, large-scale cross-cluster communication. It employs VXLAN tunnels [19] to establish basic connectivity, implements Multi-Gateway Aggregation to enhance cross-cluster throughput, and utilizes eBPF/XDP technologies [10] for efficient packet forwarding at gateways. These strategies collectively optimize the routing and processing of substantial traffic volumes across multiple Kubernetes clusters. Furthermore, this layer integrates failover mechanisms through Bucket-Based Consistent ECMP, which ensures robust traffic management during gateway failures and maintains load balancing among the cluster's

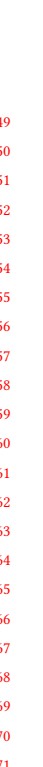

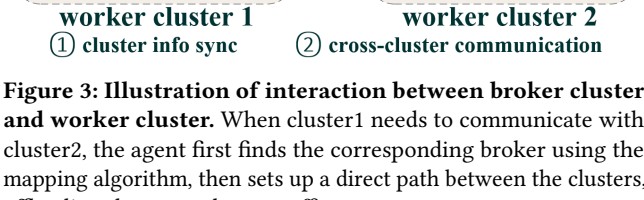

Figure 3: Illustration of interaction between broker cluster and worker cluster. When cluster1 needs to communicate with cluster2, the agent first finds the corresponding broker using the mapping algorithm, then sets up a direct path between the clusters, offloading the cross-cluster traffic.

gateways. Additionally, X-ClusterLink agents are instrumental in synchronizing cluster information, and managing routing configurations across multiple gateways.

# 4 Broker Layer Design

## 4.1 Handling Burst Traffic by Broker Cluster

We virtualize a broker cluster utilizing Kubernetes to manage traffic spikes and prevent overload, significantly enhancing scalability and flexibility beyond the capabilities of a single broker. Distributed across multiple nodes, our brokers ensure efficient cross-cluster information synchronization with minimal latency [23]. Each broker synchronizes the cross-cluster communication interfaces and ensures consistent forwarding rules [9]. This approach adeptly meets growing network demands by deploying standardized broker instances within the Kubernetes framework.

To achieve effective load balancing among brokers, we deploy an innovative mapping algorithm within the worker clusters to identify and select the appropriate broker. This algorithm dynamically recalculates the relationships between brokers and clusters upon any changes to their configurations. Detailed explanations of the mapping algorithm's operations between worker clusters and brokers will be provided in §4.2.

Figure 3 demonstrates the interaction between brokers and worker clusters. For instance, when a pod in cluster 1 needs to access a service in cluster 2, it first uses the mapping algorithm to identify the corresponding broker based on the established relationship. Following this, it synchronizes the cross-cluster communication interfaces, and a tunnel is then established between the two clusters, efficiently offloading the cross-cluster communication traffic.

## 4.2 Broker Cluster Mapping Algorithm

### 4.2.1 System Model.
In X-ClusterLink framework, the set of brokers is $B = \{b_1, b_2, \ldots, b_{|B|}\}$, with each broker $b \in B$ having a data synchronization capacity $A(b)$. Worker clusters are represented by $C = \{c_1, c_2, \ldots, c_{|C|}\}$. Complete set of Virtual Private Clouds

(VPCs) is $V = \{v_1, v_2, \ldots, v_{|V|}\}$. Each worker cluster $c \in C$ contains a subset of VPCs $V^c$, such that $V = \bigcup_{c \in C} V^c$. Tenants are represented by $T = \{t_1, t_2, \ldots, t_{|T|}\}$, with each tenant $t \in T$ associated with a subset of VPCs $V^t$, forming the complete set $V = \bigcup_{t \in T} V^t$. The traffic demand for each VPC $v \in V$ is denoted as $f(v)$.

### 4.2.2 Problem Definition.
We formally define the Broker Cluster Mapping (BCM) problem with the following constraints: (1) *Worker Cluster Constraint:* Each worker cluster is mapped to only one broker. (2) *Tenant Constraint:* Each tenant can only be mapped to a limited number of brokers to prevent a single tenant's traffic from affecting all brokers, thereby ensuring enhanced security [27].

We use a binary variable $x_v^c \in \{0, 1\}$ to denote whether a VPC $v \in V$ is mapped to a worker cluster $c \in C$. Similarly, binary variable $y_t^b \in \{0, 1\}$ represents whether the broker $b \in B$ is assigned to the VPC belonging to the tenant $t \in T$. Additionally, $z_c^b \in \{0, 1\}$ specifies whether the worker cluster $c \in C$ is mapped to a broker $b \in B$. The objective of Broker Cluster Mapping (BCM) is to achieve the optimal load-balance among all brokers, and is mathematically formulated as follows:

$$\min \lambda$$

$$S.t. \begin{cases} \sum_{b \in B} z_c^b = 1, & \forall c \in C \\ \sum_{c \in C} x_v^c = 1, & \forall v \in V \\ \sum_{c \in C} z_c^b \cdot x_v^c \leq y_t^b, & \forall v \in V^t, b \in B, t \in T \\ \sum_{c \in C} \sum_{v \in V} z_c^b \cdot x_v^c \cdot f(v) \leq \lambda A(b), & \forall b \in B \\ \sum_{b \in B} y_t^b \leq k \leq |B|, & \forall t \in T \\ x_v^c \in \{0, 1\}, & \forall v \in V, c \in C \\ y_t^b \in \{0, 1\}, & \forall t \in T, b \in B \\ z_c^b \in \{0, 1\}, & \forall c \in C, b \in B \end{cases} \quad (1)$$

### 4.2.3 Rounding-Based Mapping Algorithm.
To address the challenges in Eq. (1), we introduce the Rounding-Based Broker Cluster Mapping (RBCM) algorithm, outlined in Algorithm 1. The RBCM algorithm has two main phases:

**Phase 1: Linear Programming Relaxation (LP-BCM).** We first convert the BCM problem into a linear programming model, allowing each VPC's traffic to be fractionally split across multiple brokers. By using a solver like CPLEX [11], we obtain fractional solutions $\{\tilde{y}_t^b\}$ and $\{\tilde{z}_c^b\}$, with the optimal value $\tilde{\lambda}$.

**Phase 2: Randomized Rounding.** We then convert these fractional solutions into integers. For each tenant $t \in T$ and broker $b \in B$, $\tilde{y}_t^b$ is rounded to 1 with a probability equal to its fractional value. After determining $\{\tilde{y}_t^b\}$, we assign each tenant flow $f$ a default broker. Each decision is made independently.

Finally, we set $\tilde{z}_c^b$ to 1 with a probability $\frac{\tilde{z}_c^b}{\sum_{b \in B_c} \tilde{z}_c^b}$ for each cluster $c \in C_v$ associated with VPC $v$. If a cluster $c$ is not linked to VPC $v$, $\tilde{z}_c^b$ is set to 0.

**Algorithm 1** RBCM: Rounding-based Broker Cluster Mapping Algorithm

1: **Step 1: Solving the Relaxed BCM Problem**
2: Construct a linear programming formulation of the problem in Eq. (1)
3: Assign values to $\{\hat{x}_v^c\}$ according to the existing clusters
4: Obtain the optimal fractional solutions $\{\tilde{y}_t^b\}$ and $\{\tilde{z}_c^b\}$
5: **Step 2: Broker Cluster Selection for Each Worker Cluster**
6: **for** Each tenant $t \in T$ **do**
7:   Define $C_t = \{c|\hat{y}_t^b = 1\}$
8:   **repeat**
9:     **for** Each broker $b \in B$ **do**
10:       Set $\hat{y}_t^b = 1$, with the probability of $\tilde{y}_t^b$
11:   **until** $|C_t| > 0$
12: **for** each worker cluster $c \in C$ **do**
13:   Define $B_c = \{b|\hat{z}_c^b = 1, c \in C\}$
14:   **for** each $b \in B_c$ **do**
15:     Set $\hat{z}_c^b = 1$, with the probability of $\frac{\tilde{z}_c^b}{\sum_{b \in B_c} \tilde{z}_c^b}$

## 5 Worker Layer Design

### 5.1 Optimizing Forwarding through Multi-Gateway Aggregation and eBPF/XDP

*5.1.1 Enhancing Cluster Throughput with Multi-Gateway Aggregation.* To enhance traffic forwarding efficiency between clusters, we employ Multi-Gateway Aggregation to increase cross-cluster bandwidth. Traditional systems such as Submariner [33] typically utilize a master-slave model, where only one gateway manages traffic at any given time while others serve as backups. This model results in suboptimal resource utilization and becomes inadequate as traffic demands and cluster sizes increase.

We address the challenge of cross-cluster traffic exceeding the capacity of a single network interface by implementing a highly robust Multi-Gateway Aggregation using Open Virtual Network (OVN) [25] and Equal-Cost Multi-Path (ECMP) [12] technologies. Typically, nodes within a cluster are strategically located in a single data center where intra-cluster bandwidth substantially exceeds cross-cluster bandwidth. OVN creates a virtual overlay for centralized management [31], and ECMP facilitates the efficient hashing and even distribution of packets across multiple active gateways, preventing any one gateway from becoming a bottleneck. This Multi-Gateway Aggregation operation significantly increases total forwarding capacity and enhances the reliability of cross-cluster communication, effectively handling high traffic demands and leading to a more resilient and highly scalable network infrastructure. For specific ECMP implementations, see §5.2.

*5.1.2 Leveraging eBPF/XDP for Efficient Packet Forwarding at Gateways.* To minimize packet latency in gateway pods, we employ eBPF/XDP technology to enhance traffic forwarding. In a Kubernetes environment, each gateway pod utilizes the macvlan technique to connect its virtual network interface to the host node's physical interface. The eXpress Data Path (XDP) operates early in packet processing, bypassing the kernel's network stack to speed up traffic forwarding between the node and the gateway pod. When

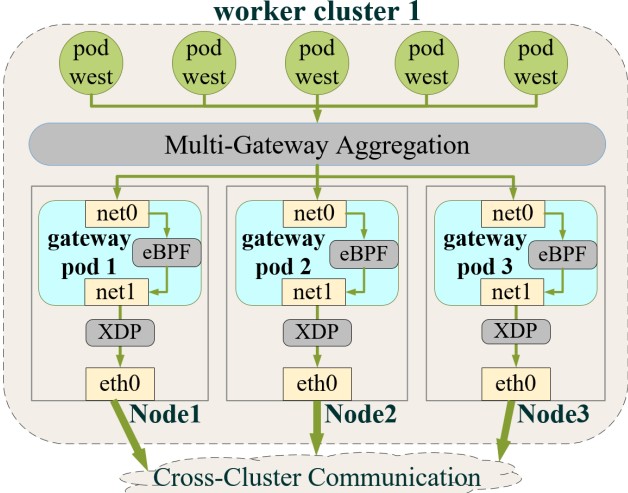

**Figure 4: Illustration of Multi-Gateway Aggregation and eBPF/XDP:** Packets for cross-cluster are first optimized by Multi-Gateway Aggregation, combining gateway bandwidth. Then, the eBPF/XDP program bypasses the kernel and sends traffic directly to the physical NIC.

cross-cluster traffic hits the node's physical interface, the XDP program quickly detects it and forwards it directly to the gateway, which then routes it to the appropriate pod.

Each gateway pod has two network interface cards (NICs): one for internal Kubernetes networks and another for external traffic via the host's physical interface. This setup ensures seamless communication between internal and external networks. We also enhance data forwarding by implementing eBPF programs at the gateway's socket layer. When an intra-cluster packet arrives, the eBPF program performs source network address translation (SNAT) and forwards the packet directly to the physical interface. For external packets, it retrieves the mapping, performs destination network address translation (DNAT), and sends the packet to the internal network. Integrating eBPF and XDP programs with the NICs significantly boosts packet forwarding speed, reducing CPU usage and enabling the gateway to handle high volumes of traffic efficiently.

*5.1.3 Comprehensive Workflow for Optimizing Forwarding with Multi-Gateway Aggregation and eBPF/XDP.* Figure 4 illustrates the traffic paths established through Multi-Gateway Aggregation and eBPF/XDP acceleration. When a cluster encounters large-scale cross-cluster traffic, since the bandwidth between nodes within the cluster significantly exceeds each node's cross-cluster bandwidth, the traffic is immediately distributed to multiple gateways for efficient cross-cluster forwarding. These gateways are strategically deployed across various nodes within the cluster to fully utilize the cross-cluster communication bandwidth.

In a gateway pod, when the net0 NIC, tasked with handling internal Kubernetes network communications, receives traffic designated for cross-cluster forwarding, an eBPF (extended Berkeley Packet Filter) program inspects the packet header. This program captures the source and destination IP addresses and executes several network functions, such as Network Address Translation (NAT).

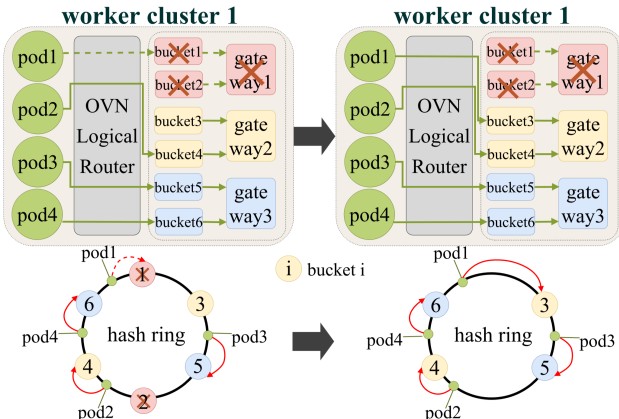

**Figure 5: Illustration of Failover by Bucket-Based Consistent ECMP:** When gateway1 fails, the OVN logic router reassign pod1, originally bound to bucket1 on gateway1, to bucket3 on gateway2 by Bucket-Based Consistent ECMP. A fixed-length hash ring minimizes global route updates, with each gateway assigned multiple buckets that are evenly distributed on the ring to maintain load balance among the gateways.

The packet is then quickly relayed to the outbound NIC, net1, with XDP (Express Data Path) technology accelerating the forwarding process from net1 to the physical NIC, eth0.

In summary, the deployment of multiple gateway pods significantly enhances the forwarding of cross-cluster traffic. By employing Multi-Gateway Aggregation and eBPF/XDP technologies, we enable multiple gateway containers to operate concurrently. This approach effectively harnesses the bandwidth of external network interfaces across various nodes within the cluster, thereby improving cross-cluster communication.

## 5.2 Failover by Bucket-Based Consistent ECMP

*5.2.1 Gateway Failure Management by Bucket-Based Consistent ECMP.* To implement Multi-Gateway Aggregation (§5.1), we utilize Equal-Cost Multi-Path (ECMP) technology [12], enabling simultaneous data forwarding across multiple gateways to enhance throughout. However, in large-scale network environments, failures, particularly at gateways, are inevitable and can significantly impact inter-cluster communication efficiency. Traditional ECMP, which relies on random hashing, necessitates a complete system reroute whenever gateways are added or removed. This process requires updating gateway assignments and routing tables for each pod, which can be disruptive and time-consuming. To address this challenge, we adopt a Bucket-Based Consistent ECMP approach.

Initially, we establish a hash ring of fixed length. Subsequently, we meticulously create multiple buckets for each gateway and hash them onto this ring. When a pod requiring cross-cluster communication initiates a request, it is efficiently hashed onto the hash ring, followed by a search for the first bucket clockwise on the ring. Once the first bucket clockwise is found, the pod is directly routed to the gateway corresponding to that bucket.

This algorithm's effectiveness stems from the fixed length of the hash ring, ensuring that each pod's hash value remains unchanged. Consequently, with our approach, adding or removing gateways

affects only the routing of pods proximal to the buckets linked with the modified gateway. This localized change circumvents the extensive global routing updates typically required by traditional ECMP, reducing downtime. Furthermore, each gateway is assigned multiple buckets, evenly distributed on the hash ring, ensuring a load-balance distribution across gateways.

Additionally, we leverage Open Virtual Network (OVN) to streamline the connections between pods and their respective gateways through virtual routers. OVN's role as a mediator in abstracting the underlying physical network complexities is crucial for seamless integration and efficient management of these connections. This architecture ensures that modifications in the calculated ECMP routing information have no detrimental impact on the pod routing tables. Consequently, it supports seamless failover, significantly enhancing system resilience by ensuring continuous service availability even during gateway changes.

*5.2.2 Workflow for Failover through Bucket-Based Consistent ECMP.* In Figure 5, we illustrate how Bucket-Based Consistent ECMP supports failover during gateway failures. Initially, cluster1 employed three gateway pods, configured via OVN logic router with Bucket-Based Consistent ECMP routing. According to the Bucket-Based Consistent algorithm, pod1 was linked to bucket1 and assigned to gateway1, while pod2 and pod3, pod4 were respectively mapped to gateway2 and gateway3. Upon detecting a failure in gateway1 via a Kubernetes probe, the Bucket-Based Consistent algorithm recalculates the pod-bucket bindings, reassigning pod1 to gateway2 while maintaining other mappings to avoid unnecessary hash recalculations and global routing disruptions. Once gateway1 is restored, the algorithm reverts pod1 to it, preserving the rest of the mappings. This process effectively demonstrates that Bucket-Based Consistent ECMP minimizes the impact of failures on global routing and ensures balanced load distribution across gateways.

In summary, our multi-gateway pod architecture employs Bucket-Based Consistent ECMP to strategically minimize the impact of gateway changes. It achieves this by remapping only the pods associated with the affected gateway, rather than the entire system. This targeted approach not only drastically reduces the time required for failover and recovery but also ensures even load distribution by mapping multiple buckets to each gateway. Moreover, this strategy facilitates the rapid scalability of gateways, thereby enhancing cross-cluster communication.

## 6 Performance Evaluation

## 6.1 Performance Metrics and Benchmarks

*6.1.1 Performance Metrics:* We categorize the performance metrics into three sets:

(1) To illustrate the high efficiency of X-ClusterLink in cross-cluster communication, we adopt the following performance metrics: (i) CPU utilization of gateways; (ii) maximum forwarding load of gateways; (iii) round-trip time (RTT); (iv) packet loss rate; (v) request per second; (vi) time per request.
(2) To evaluate the failover capability of X-ClusterLink during abnormal events, we measure (i) recovery latency of abnormal events.

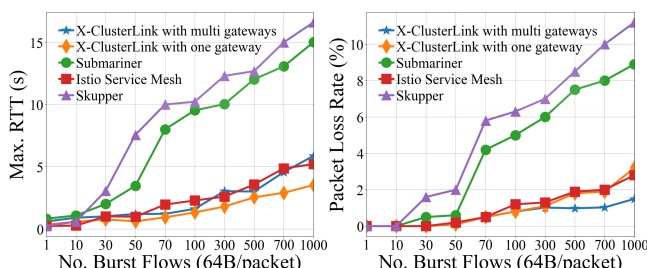

**Figure 6: Max. RTT vs. No. Burst Flows**

**Figure 7: Packet Loss Rate vs. No. Burst Flows**

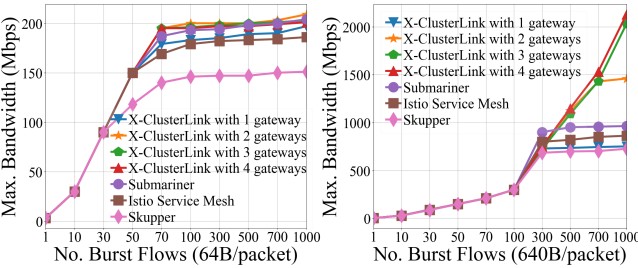

**Figure 10: Max. Bandwidth vs. Burst Flows (64B/packet)**

**Figure 11: Max. Bandwidth vs. Burst Flows (640B/packet)**

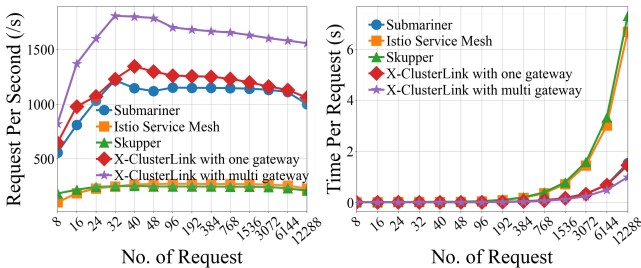

**Figure 8: Request Per Second vs. No. Request**

**Figure 9: Time Per Request vs. No. Request**

(3) To show the advantages of the X-ClusterLink during cluster scaling, we adopt the following metrics in large-scale tests: (i) cross-cluster information synchronizes latency; and (ii) CPU load of brokers.

*6.1.2 Benchmarks:* In comprehensive testbed experiments, we evaluate the efficiency of X-ClusterLink with the designs of Multi-Gateway Aggregation and eBPF/XDP and assess the robustness of X-ClusterLink with the designs of Bucket-Based Consistent ECMP and broker cluster. We then compare the performance of X-ClusterLink against that of other typical fabrics. The first fabric is Istio Service Mesh [13], the most popular choice in service meshes. Istio provides a robust way to manage microservices and their interactions and is ideally suitable for use on Kubernetes clusters. The second fabric is Skupper [32], a Layer-7 service facilitating multi-cluster interconnection. Skupper enables secure communication across Kubernetes clusters by establishing an ad-hoc virtual networking substrate. The third fabric is Submariner [33], which allows direct networking between pods and services in different Kubernetes clusters, whether on-premises or in the cloud. Additionally, X-ClusterLink functionality is demonstrated in two scenarios: single gateway pod and multiple gateway pods.

*6.1.3 System Implementation:* We establish a testbed consisting of 10 servers, all running Ubuntu 18.04 with Linux kernel 5.4. These servers are equipped with a 22-core Intel Xeon 6152 processor, 128GB of memory, and an Intel X710 10GbE NIC. We deploy 30 Kernel-based VMs (KVMs) [29] across 5 of these servers, with each KVM node configured with 4 vCPUs and 6GB of memory. For the X-ClusterLink architecture, we organize these 30 KVMs into 6 evenly distributed Kubernetes worker clusters.

Connectivity between each KVM working node and its host machine is seamlessly facilitated through a Linux bridge, allowing all KVMs on different host machines to function cohesively as a single Kubernetes cluster.

## 6.2 Efficiency Evaluation

*6.2.1 Coping with Large Traffic:* To test the system's capacity to handle large-scale traffic effectively, we initiate $10^3 - 10^4$ burst flow events, with traffic intensity ranging from 1Mbps to 1Gbps [2]. Source and destination clusters for each flow are selected randomly. We strategically utilize multiple iperf client pods within these clusters to generate burst flows, setting packet sizes to 64 Bytes to rigorously test the forwarding performance of X-ClusterLink [22]. This setup allows for precise measurements of critical performance metrics such as RTT and bandwidth.

Additionally, we configure eight pods within a single cluster to offer HTTP services, which are aggregated and exposed through a Kubernetes Service to enable load balancing. Using the Apache Bench tool located in a separate cluster, we conduct a 30-second concurrent access test to assess service responsiveness. This test provides valuable data on Requests per Second (RPS) and Time Per Request, showcasing the system's performance under varying levels of service access concurrency.

Figures 6-9 illustrate the performance of X-ClusterLink under high-traffic conditions, focusing on maximum RTT, packet loss rate, Requests Per Second (RPS), and Time Per Request (TPR).

Figure 6 illustrates that X-ClusterLink significantly lowers the maximum Round-Trip Time (RTT) compared to other network fabrics. Specifically, under conditions of 1K burst traffic, it reduces the maximum RTT by 71.7% relative to Submariner and by 68.7% relative to Skupper. Additionally, Figure 7 highlights X-ClusterLink's superior performance in reducing packet loss rates, achieving reductions of 73.6% compared to Submariner and 79.0% compared to Skupper in a multi-Kubernetes cluster cloud environment. These results demonstrate X-ClusterLink's efficiency in maintaining high-quality network performance under substantial load conditions.

Figure 8 demonstrates that X-ClusterLink substantially outperforms other network fabrics in Requests Per Second (RPS), handling up to 55.7%, 575.4%, and 638.9% more requests than Submariner, Istio Service Mesh, and Skupper, respectively, under scenarios of 10K+ concurrent access requests. Concurrently, Figure 9 clearly shows that X-ClusterLink significantly reduces Time Per Request (TPR), achieving reductions of 35.8%, 85.1%, and 86.4% compared to the same competitors. These remarkable enhancements in handling

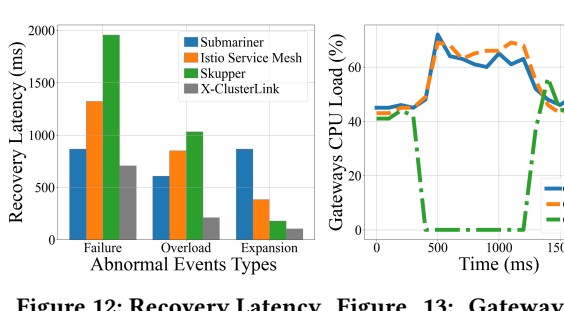

**Figure 12: Recovery Latency vs. Abnormal Events**

**Figure 13: Gateways CPU Load in a Cluster vs. Time**

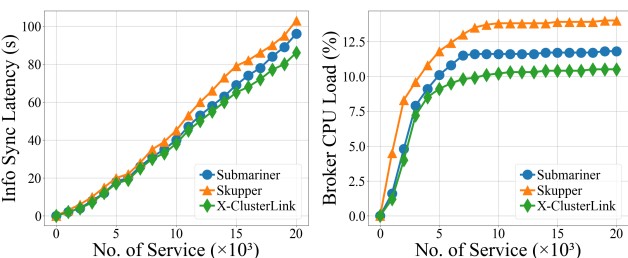

**Figure 14: Info Sync Latency vs. No. Service**

**Figure 15: Broker CPU Load vs. No. Service**

HTTP requests and processing efficiency are largely attributed to the highly effective use of Multi-Gateway Aggregation and bucket-based ECMP, which optimally distributes burst traffic loads across multiple gateways, thereby optimizing network throughput and minimizing response times.

To accurately assess the performance enhancement provided by Multi-Gateway Aggregation configurations, we conduct burst traffic tests using iperf with packet sizes of 64 bytes and 640 bytes, detailed in Figure 10 and Figure 11. For 64-byte packets, all tested systems, including Submariner, Istio Service Mesh, Skupper, and X-ClusterLink, show a bandwidth of approximately 200 Mbit/s, with no significant edge for X-ClusterLink due to the iperf pod's generation limit. However, with 640-byte packets, X-ClusterLink's multi-gateway setup significantly outperforms, achieving up to 2130 Mbit/s by more effectively utilizing the combined bandwidths of multiple nodes' physical network interfaces, unlike the single-gateway configurations which were constrained by the bandwidth limits of individual network cards.

## 6.3 Robustness Evaluation

*6.3.1 Coping with Abnormal Events:* We evaluate the robustness of X-ClusterLink in handling abnormal events such as gateway failures, overloads, and expansions, focusing particularly on the recovery time metric. X-ClusterLink employs Multi-Gateway with Bucket-Based Consistent ECMP that facilitates rapid and efficient failover in these scenarios.

Compared to X-ClusterLink, other network solutions have varied recovery strategies for abnormal events. Submariner utilizes a standby gateway node that activates upon the primary gateway's failure, providing a quicker but still conditional response. Istio Service Mesh, with only one gateway per cluster, faces delays as it waits for the gateway to recover. Skupper's recovery time is constrained by the speed of its Layer 7 routers' recovery processes.

Figures 12-13 demonstrate the capability of X-ClusterLink's Bucket-Based Consistent ECMP in managing abnormal events like gateway failures and expansions. To assess recovery latency, we transmit 2000 Ping probe packets per second across clusters through the gateways experiencing disruptions. Recovery latency is gauged by the number of probe packets lost during these events.

The results shown in Figure 12 reveal that X-ClusterLink significantly reduces recovery latency compared to other network architectures. Specifically, X-ClusterLink's gateway failure recovery latency is 706 milliseconds, approximately 1.89 times faster than that of Istio Service Mesh and 2.77 times faster than Skupper.

Figure 13 illustrates the load dynamics during a gateway failure within an X-ClusterLink cluster configured with three gateways. Initially, each gateway operates at around 45% CPU load. Following the failure of gateway pod3 at 400ms, which stops forwarding traffic, the system redirects the load to pod1 and pod2, resulting in increased CPU utilization at these nodes. By 1200ms, pod3 restarts successfully, and the load distribution among the three gateways stabilizes back to normal levels.

*6.3.2 Coping with Cluster Scaling:* Based on the designs of the X-ClusterLink framework, we implement a broker cluster by Kubernetes and propose the Rounding-Based Broker Cluster Mapping (RBCM) algorithm, aimed at enhancing load balancing within broker clusters and reducing latency in cross-cluster information synchronization. This section outlines scalability tests conducted to assess the efficiency of the Broker Layer.

We conduct a testbed to compare cross-cluster information synchronization latency and agent overhead among X-ClusterLink, Submariner, and Skupper. As depicted in Figures 14-15, X-ClusterLink consistently demonstrates lower latency and overhead as the number of services increases. Notably, for $2 \times 10^4$ services, X-ClusterLink achieves a service synchronization latency of 86 seconds, which is 19.8% faster than Skupper (103 seconds) and 11.6% faster than Submariner (96 seconds). Additionally, X-ClusterLink exhibits a 33.5% reduction in synchronization overhead compared to Skupper and a 12.4% reduction compared to Submariner.

X-ClusterLink leverages its broker cluster and the RBCM algorithm to efficiently map worker clusters to different brokers, thus enhancing information synchronization. This strategy minimizes latency and overhead associated with agent updates and synchronization, effectively optimizing cross-cluster information exchange during cluster expansion.

## 7 Conclusion

This paper introduces X-ClusterLink, a framework designed to efficiently enhance communication across multi-Kubernetes clusters. X-ClusterLink optimizes cross-cluster communication using Multi-Gateway Aggregation, eBPF/XDP, and the broker cluster to manage burst traffic efficiently. Additionally, it integrates Bucket-Based Consistent ECMP to handle network anomalies robustly. Our comprehensive testbed experiments and simulations demonstrate significant improvements in latency and throughput, affirming X-ClusterLink as a superior solution for application virtualization and cross-cluster communication.

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

# A Performance Analysis of Broker Cluster Mapping Algorithm

*Theorem 1*: The BCM problem is NP-hard.

*Proof*: Consider a simplified version of the BCG problem without the *Tenant Constraint*. This simplification reduces the BCG problem to a Parallel Machine Scheduling (PMS) problem[20], known as NP-hard. Thus, the BCG problem is also NP-hard. Due to the space limit, we omit the detailed proof here.

This section provides an approximate analysis of the tenant constraint ratio and load balancing performance. We first introduce two well-known probability theory lemmas:

*Lemma 2 (Chernoff Bound)*: Given $n$ independent variables: $x_1, x_2, \ldots, x_n$, where $x_i \in [0, 1]$. Let $\mu = \mathbb{E}[\sum_{i=1}^{n} x_i]$. Then,

$$\Pr\left[\sum_{i=1}^{n} x_i \geq (1+\epsilon)\mu\right] \leq e^{-\frac{\epsilon^2 \mu}{2+\epsilon}}, \tag{2}$$

where $\epsilon$ is an arbitrarily positive value.

*Lemma 3 (Union Bound)*: Given a countable set of $n$ events $A_1, A_2, \ldots, A_n$, each event $A_i$ happens with possibility $\Pr(A_i)$. Then,

$$\Pr\left[\bigcup_{i=1}^{n} A_i\right] \leq \sum_{i=1}^{n} \Pr(A_i). \tag{3}$$

**Analysis of the worker Cluster Constraint**: The expected number of worker clusters assigned to the flow $f$ is:

$$\mathbb{E}\left[\sum_{c \in C} \tilde{z}_c^b\right] = \sum_{c \in C} \mathbb{E}[\tilde{z}_c^b] = \sum_{c \in C} \Pr[\tilde{z}_c^b = 1]$$
$$= \sum_{c \in C_v} \Pr[\tilde{z}_c^b = 1] + \sum_{c \in C \backslash C_v} \Pr[\tilde{z}_c^b = 1] \tag{4}$$
$$= 1 + 0 = 1$$

where the last equation holds according to the second step of RBCM, which selects a default cluster from $C_v$ for flow $f \in F$ with the probability of $\frac{\tilde{z}_c^b}{\tilde{x}_v^c}$, thus, the RBCM algorithm will assign only one worker cluster to each flow, ensuring compliance with the flow constraint.

**Analysis of the Tenant Constraint**: The first step of RBCM will yield the fractional solution $\tilde{y}_t^b$ of the relaxed BCM problem. Using randomized rounding, $\tilde{y}_t^b$ is set to 1 with the probability of $\tilde{y}_t^b$.

Thus, the expected number of worker clusters allocated to the VPC $v$ is given by:

$$\mathbb{E}\left[\sum_{c \in C} \tilde{y}_t^b\right] = \sum_{c \in C} \tilde{y}_t^b \leq k \tag{5}$$

*Theorem 4*: With the rounding-based mapping algorithm, the number of brokers assigned to the tenant $t$ will not exceed $k$ by a factor of $3 \ln d + 3$ with high probability, where $d$ represents the number of VPCs.

*Proof*: For each $v_c$, $\tilde{y}_t^b \in \{0, 1\}$ are independent variables with an expected value $\mathbb{E}\left[\sum_{c \in C} \tilde{y}_t^b\right] \leq k$. According to Lemma 2, we have:

$$\Pr\left[\sum_{c \in C} v_c \geq (1+\epsilon)k\right] \leq e^{-\frac{\epsilon^2 k}{2+\epsilon}} \tag{6}$$

We assume that

$$e^{-\frac{\epsilon^2 k}{2+\epsilon}} \leq \frac{1}{d^3}, \ d = |T| \tag{7}$$

which implies that the probability bound in Eq. 4 rapidly approaches zero as the number of VPCs $d$ increases. To satisfy this, $\epsilon$ should be:

$$\epsilon \geq \frac{3 \ln d + \sqrt{9 \ln^2 d + 24k \ln d}}{2k} \tag{8}$$

If we select $\epsilon = \frac{3 \ln d}{k} + 2$, the above inequality holds. In other words, we have:

$$\Pr\left[\sum_{c \in C} \tilde{y}_t^b \geq (1+\epsilon)k\right] \leq \frac{1}{d^3}, \ \epsilon = \frac{3 \ln d}{k} + 2 \tag{9}$$

Finally, we ensure the upper bound on the probability the number of worker clusters assigned to a VPC is violated by Lemma 3:

$$\Pr\left[\bigcup_{v \in V} \sum_{c \in C} \tilde{y}_t^b \geq (1+\epsilon)k\right]$$
$$\leq \sum_{v \in V} \Pr\left[\sum_{c \in C} \tilde{y}_t^b \geq (1+\epsilon)k\right] \tag{10}$$
$$\leq d \cdot \frac{1}{d^3} = \frac{1}{d^2}, \ \epsilon = \frac{3 \ln d}{k} + 2$$

Therefore, the number of brokers assigned to tenant $t$ will not exceed $k$ by a factor of $1 + \epsilon = \frac{3 \ln d}{k} + 3$ with high probability.

**Load Balancing Performance Analysis**: We calculate the expected forwarding load of worker clusters and bound the probability that the load will be violated. First, we define $l_f^c$ as the forwarding load of the cluster $c \in C$ assigned to flow $f \in F$:

$$l_f^c = \begin{cases} t(f), & \text{with the probability of } \frac{\tilde{z}_c^b}{\tilde{z}_v^c} \\ 0, & \text{otherwise} \end{cases} \tag{11}$$

The expected forwarding load of the cluster $c$ is:

$$\mathbb{E}\left[\sum_{f \in F^r} l_f^c\right] = \sum_{f \in F^r} \mathbb{E}[l_f^c] = \sum_{f \in F^r} t(f) \cdot z_f^c \leq \lambda \cdot A(b) \tag{12}$$

To understand the relationship between load variables and the optimal result, we define:

$$\beta = \frac{\lambda \cdot A(b)}{\max_{f \in F^r} t(f)} \tag{13}$$

*Theorem 5*: The rounding-based mapping algorithm achieves a load balancing factor at most $\frac{3 \ln d}{\beta} + 3$ times worse than the optimal result with high probability. Due to the space limit, we omit the detailed proof here.

