# OpenReview forum: "X-ClusterLink: An Efficient Cross-Cluster Communication Framework in Multi-Kubernetes Clusters"
_ACM.org/TheWebConf/2025/Conference — WWW 2025 Poster_

### Official Review · Reviewer_rEZu · 2024-11-26

**Novelty:** 5
**Technical Quality:** 5

**Review:**

# **Review Summary**
The paper proposes X-ClusterLink, a framework aimed at achieving efficient cross-cluster communication in multi-Kubernetes cluster environments. The framework is designed to address the critical requirements of low latency, high throughput, and strong robustness.
## **Quality**

The paper demonstrates good overall quality. It provides a thorough analysis of related work, a well-structured methodology, and detailed experimental validation with sufficient supporting details.

## **Clarity**

The paper is clearly written and well-organized, making the proposed framework and its components easy to understand. Figures and tables effectively support the textual content.

## **Originality**

The work offers a practical solution to cross-cluster communication challenges in multi-Kubernetes environments. While the contributions are relevant, a clearer emphasis on how this framework differs from existing approaches would further highlight its originality.

## **Significance**

The proposed framework addresses an important issue by focusing on low latency, high throughput, and strong robustness, making it significant for distributed systems applications.
## **Strengths**
- Detailed analysis of related work, providing a solid foundation for the proposed approach.
- Thorough experimental validation with sufficient supporting details.
- Clear and logical presentation of the system and its components.
## **Weaknesses**

While I find the overall writing and presentation of the paper to be clear and well-structured, I am less familiar with the specific technical domain addressed in this work. As a result, I may not be able to provide in-depth or highly constructive feedback on the technical contributions. A more detailed evaluation of the technical aspects may benefit from input by experts with greater specialization in this area.

**Questions:**

Q1: To what extent can the validity of the experimental results be affected by the experimental settings?

**Reviewer Confidence:**

1: The reviewer's evaluation is an educated guess

**Scope:**

4: The work is relevant to the Web and to the track, and is of broad interest to the community

---

### Official Review · Reviewer_867S · 2024-11-26

**Novelty:** 4
**Technical Quality:** 5

**Review:**

This paper discusses a newly developed cross-cluster communication framework called X-ClusterLink. Authors explain the utilised technologies and a new algorithm for mapping clusters to brokers that manage the synchronization between them.  Later they explain how they have utilized new technologies in their framework such as eBPF/XDP and Bucket-Based ECMP. They conclude with comparing their framework to the existing cross-cluster networking solutions.

It is a well-written paper with a clear explanation of the limitations of existing solutions and the design decisions taken to design the framework. Authors have implemented a well-thought algorithm, that tackles cross-cluster communication problems with existing technologies, however it is difficult to say that they have used a novel approach. It seems that, they have built a system that utilises existing technologies in an efficient way. Their experiment setup and experiment methodologies are explained clearly where they highlight their solution’s superiority to the existing frameworks clearly. It’s also a nice contribution that they analyse the performance of their algorithm analytically.

It would be nice to see if the outcomes of the proofs were highlighted in the result section. Even though, the design decisions are backed up by the formulas and algorithm explanation, there are no mentions of them in the results section. The authors can mention that the experiment results show that their proof holds and optimality of the algorithm can be seen from the plots. Another missing point is the analysis of the algorithm’s performance with varying cross cluster network delays, which might be quite relevant for cross-cluster application that are deployed in various geographies. The experiment setup is quite ideal in terms of this, as all the clusters are essentially located at the same place. It would be interesting to see how failure detections would work in variable network delays.

 Experiments seem a bit too ideal. It would be nice to see a more dynamic environment, maybe some tenants and clusters joining in and out, that would allow readers to see how the Broker Cluster Mapping Algorithm behave dynamically (it raises a question since it is used by a solver which might cause significant overheads) and a more realistic application, where inter-service communication is more sporadic. Highlighting the limitations of the current work and future research directions would also be valuable.

Overall, it is a valuable paper that have integrated many existing technologies into a cluster communication framework and explained the taken design decisions clearly. A new algorithm was designed and implemented to assign clusters to brokers which results in more efficient and robust cross-cluster networking. However, there is doubt that the paper really falls into the scope of the conference. The paper doesn’t provide a solution to a problem related to Web, but to a problem that can benefit some applications that are served in the Web, which is a weak connection to the scope of the conference.

Pros:
- A valuable developed framework that utilises many modern networking technologies
- Good analytical analysis of the developed algorithm
- Clear explanation and comparison with the existing solutions

Cons:
- Analytical results are not at all mentioned in the results section
- Experiments do not incorporate varying networking conditions, which are quite relevant for cross-cluster communication
- Weak connection to the scope of the conference

**Questions:**

- Do you have any experimental results which show that the proofs hold?
- Can you explain to what extent this algorithm would work in more dynamic environments?
- Did you experiment with varying and large networking delays, which are common in geographical distributed clusters?

**Reviewer Confidence:**

2: The reviewer is willing to defend the evaluation, but it is likely that the reviewer did not understand parts of the paper

**Scope:**

2: The connection to the Web is incidental, e.g., use of Web data or API

---

### Official Review · Reviewer_AX6Q · 2024-11-28

**Novelty:** 5
**Technical Quality:** 6

**Review:**

In this work, the authors provide X-ClusterLink, a framework designed for efficient cross-cluster communication in multi-Kubernetes clusters. Cross-cluster commnunication is important for the quality of Web services. Through the broker-worker architecture, X-ClusterLink mainly focuses on low latency, high throughput, strong robustness and scalability of communication. The paper is well organized and the experiments can demonstrate most of the advantages. The pros and cons are listed as follows:

Pros:
1. The paper focuses on an important problem of web services, the cross-cluste communication problem, and the solution is  solid and feasible.
2. Experiments and the experimental analysis cover most of the advantages of the proposed solution.
3. The paper is good writing and easy to understand.

Cons:
1. What is the specific challenges of multi-k8s clusters communication compared with the work of Zeta[1][2], a scalable and robust east-west forwarding framework with gateway clusters for hyperscale clouds. It seems that Zeta also utilizes a similar broker-worker arch, and the broker cluster mapping problem are solved in a quite similar technique.  Hence, I think analysis of the differences between X-ClusterLink and Zeta is necessary.
2. I noticed that submariner, skupper and istio are all public avaiable projects, does it mean that the solution of multi-k8s clusters communication proposed in this paper is the first work to improve these fabrics?
3. What is the time consumption of the algorithm for solving the BCM problem, can the authors give a simple analysis of the time consumption? due to it is important for the final commnunication latency.
4. It is better to provide more implementation details such as code lines.
5. Small bugs: In the experimental figures, the color of different solutions are not consistent across adjacent figures, sometimes may confuse the readers.

[1] Scalable and Robust East-West Forwarding Framework for Hyperscale Clouds, ToN 2023;
[2] Zeta: A Scalable and Robust East-West Communication Framework in Large-Scale Clouds, NSDI 2022

**Questions:**

I am looking forward to discussing the following questions with the authors:

1. What is the specific challenges of multi-k8s clusters communication compared with the work of Zeta[1][2], a scalable and robust east-west forwarding framework with gateway clusters for hyperscale clouds. It seems that Zeta also utilizes a similar broker-worker arch, and the broker cluster mapping problem are solved in a quite similar technique.  Hence, I think analysis of the differences between X-ClusterLink and Zeta is necessary.
2. I noticed that submariner, skupper and istio are all public avaiable projects, does it mean that the solution of multi-k8s clusters communication proposed in this paper is the first work to improve these fabrics?
3. What is the time consumption of the algorithm for solving the BCM problem, can the authors give a simple analysis of the time consumption? due to it is important for the final commnunication latency.

**Reviewer Confidence:**

3: The reviewer is confident but not certain that the evaluation is correct

**Scope:**

4: The work is relevant to the Web and to the track, and is of broad interest to the community

---

### Official Review · Reviewer_YqFo · 2024-12-02

**Novelty:** 5
**Technical Quality:** 6

**Review:**

Summary:
This paper presents a framework designed to address the challenges of cross-cluster communication in multi-Kubernetes cluster environments. The authors evaluate the framework’s performance through experiments focused on efficiency, failover capability, and scalability during cluster expansion. Results demonstrate that the framework effectively reduces cross-cluster communication latency and improves overall throughput.

Strong points
S1. The paper enhances existing cross-cluster communication solutions by introducing a broker layer and a worker layer, which improve the efficiency of traffic forwarding and cross-cluster communication processes.
S2. Compared to the traditional ECMP algorithm, the novel Bucket-Based ECMP proposed here requires modifications only for pods associated with an affected gateway during gateway changes, thereby enhancing system stability.
S3. The authors provide a thorough and rigorous analysis of the algorithm, demonstrating its reliability and stability in terms of resource allocation and load balancing.

Weak points
W1. The termination condition in the pseudocode of Algorithm 1 is not clearly articulated and does not align fully with the process described in Section 4.2.3.
W2. The description of the failover workflow using Bucket-Based Consistent ECMP lacks details on how pod-bucket bindings are reassigned in the event of failures.
W3. The benchmarks used for comparison in this paper are largely based on methods developed over five years ago, which may limit the relevance of these comparisons. Including more recent methods from the past three years would enhance the evaluation.
W4. Figure 10 suggests that for smaller packets (64-byte), multiple gateways do not perform as well as a single gateway, indicating that the number of gateways is a sensitive parameter in this approach. Additional experiments exploring the impact of this parameter would be valuable.
W5. There are some anonymous Git websites. I want to see the codes (including the system codes and the experimental codes) during the rebuttal phase.

**Questions:**

Please address W1~W5

**Reviewer Confidence:**

3: The reviewer is confident but not certain that the evaluation is correct

**Scope:**

4: The work is relevant to the Web and to the track, and is of broad interest to the community

---

### Official Review · Reviewer_SZfc · 2024-12-02

**Novelty:** 4
**Technical Quality:** 6

**Review:**

The authors introduce X-ClusterLink, a framework for cross-cluster communication in multi-Kubernetes clusters. Considering the wide adoption of cloud-native applications and services, the use of Kubernetes clusters is a common standard and there exist multiple projects/frameworks that enhance the main functionalities of Kubernetes.

The core idea behind motivating the need for X-ClusterLink is the need for cross-cluster communication, which is required when deployed services are composed of entities that are deployed in different clusters.

Overall, the paper is well-written, and the problem is well-motivated. My main concern is whether the problem actually exists because, in my experience, cross-cluster communication is limited in cloud-native applications, and if two pods interact a lot, they are placed close to each other. What is missing from the paper is evidence regarding the need for the four design goals the authors adopt when designing their framework.

Also, as the authors mentioned in lines 79 and 80, "standard Kubernetes 79 cluster only supports up to 5,000 nodes and 150,000 pods", so it would be very useful to have at least one motivating example for a case where existing solutions are not sufficient.

**Questions:**

Have you explored the case in your experiments where all 30 KVMs where part of the same Kubernetes cluster in order to measure the "ideal" performance because there will be no need for cross-cluster communication?

**Reviewer Confidence:**

3: The reviewer is confident but not certain that the evaluation is correct

**Scope:**

3: The work is somewhat relevant to the Web and to the track, and is of narrow interest to a sub-community